# The Cardiometabolic Health of African Immigrants in High-Income Countries: A Systematic Review

**DOI:** 10.3390/ijerph19137959

**Published:** 2022-06-29

**Authors:** Danielle Mensah, Oluwabunmi Ogungbe, Ruth-Alma N. Turkson-Ocran, Chioma Onuoha, Samuel Byiringiro, Nwakaego A. Nmezi, Ivy Mannoh, Elisheva Wecker, Ednah N. Madu, Yvonne Commodore-Mensah

**Affiliations:** 1College of Medicine, Drexel University, Philadelphia, PA 19129, USA; dm3546@drexel.edu; 2School of Nursing, Johns Hopkins University, Baltimore, MD 21205, USA; oogungb3@jhu.edu (O.O.); sbyirin1@jh.edu (S.B.); 3General Medicine Research, Beth Israel Deaconess Medical Center, Boston, MA 02215, USA; rturkson@bidmc.harvard.edu; 4School of Medicine, University of California, San Francisco, CA 94143, USA; chioma.onuoha@ucsf.edu; 5Department of Physical Medicine and Rehabilitation, School of Medicine, Johns Hopkins University, Baltimore, MD 21201, USA; nnmezi1@jhmi.edu; 6School of Medicine, Johns Hopkins University, Baltimore, MD 21201, USA; imannoh1@jhmi.edu (I.M.); ewecker1@jhmi.edu (E.W.); 7College of Nursing and Public Health, Adelphi University, Garden City, NY 11530, USA; emadu@adelphi.edu; 8Johns Hopkins Bloomberg School of Public Health, Baltimore, MD 21205, USA

**Keywords:** cardiovascular risk factors, African ancestry group, immigrants

## Abstract

In recent decades, the number of African immigrants in high-income countries (HICs) has increased significantly. However, the cardiometabolic health of this population remains poorly examined. Thus, we conducted a systematic review to examine the prevalence of cardiometabolic risk factors among sub-Saharan African immigrants residing in HICs. Studies were identified through searches in electronic databases including PubMed, Embase, CINAHL, Cochrane, Scopus, and Web of Science up to July 2021. Data on the prevalence of cardiometabolic risk factors were extracted and synthesized in a narrative format, and a meta-analysis of pooled proportions was also conducted. Of 8655 unique records, 35 articles that reported data on the specific African countries of origin of African immigrants were included in the review. We observed heterogeneity in the burden of cardiometabolic risk factors by African country of origin and HIC. The most prevalent risk factors were hypertension (27%, range: 6–55%), overweight/obesity (59%, range: 13–91%), and dyslipidemia (29%, range: 11–77.2%). The pooled prevalence of diabetes was 11% (range: 5–17%), and 7% (range: 0.7–14.8%) for smoking. Few studies examined kidney disease, hyperlipidemia, and diagnosed cardiometabolic disease. Policy changes and effective interventions are needed to improve the cardiometabolic health of African immigrants, improve care access and utilization, and advance health equity.

## 1. Introduction

Cardiometabolic disease (CMD), including cardiovascular disease (CVD), diabetes mellitus, and chronic kidney disease, is the leading cause of death globally [1]. Risk factors of CMD include hypertension, obesity, and poor lifestyle including physical inactivity and smoking. The prevalence of CMD and associated risk factors in low- and middle-income countries (LMICs) has risen dramatically in the recent decades due to urbanization and the epidemiologic transition and mirrors trends in high-income countries (HICs) [2]. The differences in the prevalence of these CMD and risk factors are influenced by social determinants of health (SDoH) such as education, income, immigration status, acculturation, and length of residence in the new country.

According to the United Nations, international immigrants made up 15% of the population in HICs in 2020 [3]. The rate of migration from sub-Saharan Africa (SSA) to HICs in Europe, North America, and Oceania has increased in the post-colonial era. Over the last century, the number of Africans migrating from LMICs to HICs increased, growing from ~2 million to ~9 million between 1960 and 2000 [4]. While much of the health research focused on African populations has focused on communicable diseases such as HIV, research on non-communicable diseases (NCDs) such as CMD has received growing attention. This shift can be attributed to the global rise of NCDs in general and, particularly, across the African continent [5].

This research has also evolved to include a focus on how the process of migration impacts the health of Africans who migrate to HICs. While the general body of research exploring the impact of migration on health has yielded mixed results, some of the available research suggests a negative association between health and migration [6]. Research examining the cardiometabolic health of African immigrants residing in HICs is limited but growing. Factors associated with the change in the cardiometabolic health profile of African immigrants in HICs have been related to lifestyles and behavioral changes (e.g., changes in diet and physical activity levels), differences in access to medical care, and acculturation [6]. The purpose of this systematic review was to examine the prevalence of CMD risk factors among African immigrants who reside in HICs.

## 2. Materials and Methods

### 2.1. Search Strategy and Selection Criteria

Recommendations from the Preferred Reporting Items for Systematic Reviews and Meta-Analysis (PRISMA) [7] were used to conduct a systematic review of the cardiometabolic health of African immigrants in HICs (Appendix A). An informationist (EW) assisted with completing a detailed literature search. A search strategy was developed using keywords, controlled vocabularies (Medical Subject Headings [MeSH], Emtree, etc.), and synonyms related to SSA African immigrants and cardiometabolic health (Appendix A). The following databases were searched for this review: PubMed, Embase.com, CINAHL Plus, Cochrane Library via Wiley, Scopus, and Web of Science. Final searches covered the period of database inception till 19 July 2021. The final search strategy can be found in the Appendix A. Ethical approval was not required for this review, and the protocol was registered in the PROSPERO database CRD42021264643.

### 2.2. Study Selection and Eligibility

Articles identified from the search strategy were first imported into the EndNote reference manager [8] for deduplication. Then, the unique records were imported to Covidence, [9] a screening and data extraction tool. Initial title and abstract screening were performed by nine reviewers (OO, NN, SB, RATO, IM, CO, DM, EM, YCM) in Covidence. Two reviewers independently reviewed each article to determine eligibility for the systematic review. Any discrepancies yielded from the title, and abstract screening were resolved through discussion. Final resolutions were achieved through the help of a third reviewer who did not perform the initial screening. All eligible studies from the initial screening were then screened for full-text eligibility.

Two reviewers worked in pairs to independently screen the articles for full-text eligibility (OO, NN, SB, RATO, IM, CO, DM, EM). For each article, at least two primary reviewers and a third reviewer (YCM) served as an adjudicator in cases of discrepancies. The following inclusion criteria were used for the full-text review: (1) full text and peer-reviewed journal articles written in English or French; (2) study type: observational studies, cross-sectional studies, and cohort studies; (3) population: African immigrants aged 18 years or more; (4) setting: persons living in a country with a gross national income (GNI) of 12,695 per capita/HICs per the World Bank designation (https://datatopics.worldbank.org/world-development-indicators/the-world-by-income-and-region.html, accessed on 20 June 2022); (5) cardiometabolic risk factors or diseases: hypertension, diabetes mellitus, dyslipidemia, coronary heart disease, hypercholesterolemia, overweight/obesity, stroke, kidney/renal disease, smoking, or myocardial infarction. Non-peer-reviewed articles, abstracts only, editorials, and qualitative studies were excluded from this review. Additionally, studies that aggregated the prevalence of CMD risk factors in African immigrants, non-SSA countries (e.g., Morocco, Egypt), or other immigrant populations (e.g., Asian immigrants, Hispanic immigrants), and studies on African Americans and children were also excluded. Geographical classifications of studies were based on the United Nations Department of Economic and Social Affairs classifications [10].

### 2.3. Data Extraction

After the final studies from the full-text review were selected, data was independently extracted by all nine reviewers. A piloted data extraction form was used in Covidence for data extraction. Data on the prevalence of cardiometabolic risk factors and diseases—hypertension, diabetes, overweight/obesity, dyslipidemia, CVD, kidney disease, stroke, and tobacco use were extracted. This included data on first author and publication year, study design, population description and sample size (total sample size and African immigrant sample size), comparison population, host country, African country of origin, study eligibility criteria, recruitment method, average age, gender, and study setting. The primary outcome of interest for this review was the prevalence of CMD risk factors and diseases reported among SSA African immigrants living in HICs. The full-text review was followed similarly to the abstract/title review. Eight Reviewers (OO, NN, SB, RATO, IM, CO, DM, EM) independently voted on whether the study was eligible based on the above criteria, and a ninth reviewer (YCM) voted on the consensus. After the final studies from the full-text review were selected, data was independently extracted by all nine reviewers. Extracted data, particularly outcome measures, were cross-checked by 5 authors (DM, SB, IM, CO, RATO, OO), and discrepancies were adjudicated by consensus. The final adjudicator reviewed all inconsistencies and assisted with making a final decision (YCM).

### 2.4. Quality Assessment

Study quality was assessed using the National Institute of Health quality assessment tool for observational cohort and cross-sectional studies [11]. Domains assessed were bias due to: research question, study population, sample size justification, exposure and outcome measurements, follow-up, and assessment for confounding. Each item was initially rated ‘yes’, ‘no’, ‘not applicable’, ‘can’t determine’, or ‘not reported’, and the overall quality rating was rated as good, fair, or poor. Risk of bias plots were created using *robvis*, a web-based bias visualization [12] application, and ratings were converted into the following judgments as appropriate: high, low, unclear, moderate, critical, or no information.

### 2.5. Data Synthesis and Analysis

Study characteristics and extracted sociodemographic data were summarized in tables. Reported subgroup proportions by gender and African country of origin were synthesized. The PRISMA checklist [13] guided the reporting and ensured transparent reporting (Appendix A). For the statistical analysis, proportions of CMD risk factors and diseases were calculated, and a random effect model was used to assess the prevalence of hypertension, diabetes, dyslipidemia, overweight/obesity, and tobacco use among sub-Saharan African immigrants. Overall pooled estimates for each condition and exact 95% confidence intervals were calculated and displayed in forest plots. Data were pooled using the “*metaprop*” package in Stata/IC 16 [14] and quality assessments were summarized using the *robvis* visualization tool [12].

## 3. Results

We identified 17,951 records from our search; after removing 9296 duplicates, we screened 8655 for title and abstract eligibility, from which 8329 were excluded leaving 326 potentially eligible records (Figure 1). Full texts of 220 articles were assessed, and 185 were excluded. Thus, 35 articles were included in this review. Following quality assessment (Appendix A), overall, possible substantial bias was rated as low in 73% of the articles. All articles provided a clear objective statement, and 95% clearly defined the study population. In about 75% of the articles, possible bias due to sample size justification was rated high, unclear, or no information, while 53%, 55%, and 75% of the articles had low ratings for bias due to exposure, outcome measures, and possible confounding, respectively.

### 3.1. Sociodemographic Characteristics

#### 3.1.1. African Countries of Origin

Of the papers analyzed, 77.1% (27/35) captured data on immigrants from East Africa (Table 1). Somalian immigrants were the most commonly studied group (N = 13), followed by Ethiopians (N = 5), Kenyans (N = 1), Eritreans (N = 1), Zimbabweans (N = 1), and immigrants from Somaliland (N = 1). Data on West African immigrants were captured by 51.4% (18/35) of papers, with Ghana being the country most represented (N = 10), followed by Nigeria (N = 2), Liberia (N = 1), Sierra Leone (N = 1), Senegal (N = 1), and Guinea (N = 1).

Additionally, 8.6% (3/35) of the articles included North African immigrants born in Sudan. Central African immigrants from Equatorial Guinea, the Democratic Republic of Congo, and Angola were studied in 8.6% (3/35) of articles. Some studies analyzed data from immigrants from multiple African countries. 

#### 3.1.2. High-Income Countries of Residence

The majority of studies were primarily conducted in Europe or the United States (US), with the exception of those which took place in Australia (2/35), New Zealand (1/35), and Israel (3/35) (Figure 2). Of the studies included, 60% (21/35) were conducted in Europe, while 22.9.7% (8/35) occurred in the US. All but one of the studies was conducted in an urban setting.

#### 3.1.3. Sex and Age Distribution:

The majority of the articles (87.5%; 31/35) studied African men and women, while four studied African immigrant women only, and one article included only African immigrant men. The average age of study participants differed for all studies, but the majority included African immigrants in their 30s and 40s.

#### 3.1.4. Study Design

Of the studies analyzed, 88.6% (31/35) had a cross-sectional study design, four had a cohort design, and one used a combination of a cross-sectional and cohort design.

### 3.2. Prevalence of Cardiometabolic Disease Risk Factors

The reported prevalence of CMD risk factors among African immigrants in high-income countries is presented in Appendix A.

#### 3.2.1. Hypertension Prevalence

Several (45%; 16/35) articles reported on the prevalence of hypertension (Figure 3) [16,17,18,19,20,21,22,23,24,25,26,27,28,29,30,31]. Hypertension prevalence was assessed among African immigrants from Ghana, Nigeria, Liberia, Ethiopia, Somalia, Zimbabwe, Sudan, and Kenya. The pooled prevalence of hypertension was 26% (range: 6–55%). Where stratified by gender, hypertension prevalence among men ranged from 6 to 61.6% and 14 to 51% among women. Hypertension prevalence among African immigrants varied by the host country. In Australia, hypertension prevalence ranged between 12% (Sudanese immigrants) [28] and 30% (Ghanaian immigrants) [30]. In Finland, the prevalence was 16% (Somali immigrants), [32] and 50%, 52%, and 51% among Ghanaian immigrants in The Netherlands (Amsterdam), Germany (Berlin), and United Kingdom (London), respectively [16]. In Israel, hypertension prevalence among Ethiopian immigrants was 19% [29], while in Norway, it was between 6% [22] and 9% [22,26] among Somali immigrants. In the US, the prevalence of hypertension was 6%, [31] 17%, [27], and 22% among Somali immigrants [33], 8% [31] to 30% [34] among Ethiopian immigrants, 16% among Liberian, 16% among Kenyan immigrants, [31] 35% among Zimbabwean immigrants, [23] and 35% among Nigerian immigrants [21]. (Figure 3).

#### 3.2.2. Diabetes Prevalence

Thirty-three percent (15/35) of the articles examined diabetes in the following HIC countries: The Netherlands, Germany, the UK, US, Australia, Israel, Finland, and Norway [16,23,26,27,28,29,31,32,33,34,35,36,37,38,39,40,41]. These studies examined diabetes prevalence among immigrants from Ghana, Nigeria, Liberia, Ethiopia, Zimbabwe, Angola, Burundi, the Democratic Republic of Congo, Guinea, Sierra Leone, Somalia, Sudan, Kenya, and Morocco. Except for one study [16], all the studies that examined diabetes were conducted in urban areas. The overall diabetes prevalence was 11% (Figure 4) and ranged from 4.5% (Somalians in the US) [31] to 17.4% (Ethiopians in Israel) [29]. When examined by gender, diabetes prevalence among men ranged between 9.2% (Ethiopians in Israel) [37] and 21.08 (Somalians in the US) [33] and between 11.6% (Ethiopians in Israel) [37] and 18.6% (Somalians in Finland) [32] among women. When examining diabetes prevalence by host country, the prevalence of diabetes among African immigrants in Norway was 5% (Somalians) [26], 12.8% (Ghanaians) in the UK [16], 14.6% (Ghanaians) in Germany [16], 10.4% (Ethiopians) in Israel [37], and 13.2 (Ghanaians) in The Netherlands [16].

#### 3.2.3. Overweight/Obesity Prevalence

Most (75%; 27/36) studies reported the prevalence of overweight/obesity [16,21,22,27,28,29,30,31,32,33,35,40,42,43,44,45,46,47,48,49] among African immigrants from Ghana, Nigeria, Liberia, Somalia, Morocco, Equatorial Guinea, Somaliland, Ethiopia, Zimbabwe, Sudan, Eritrea, Kenya, Senegal, and Tunisia. The overall pooled prevalence of overweight/obesity was 59% (Figure 5), and where reported, the average prevalence of overweight and obesity were 34% and 30%, respectively. Across all studies, women were more likely to have higher rates of overweight/obesity in comparison to men. For instance, the prevalence of overweight/obesity among Somali men residing in Sweden was 35.7%, and among Somali women it was 72.4% [50].

In Australia, the prevalence of overweight and obesity among Sudanese immigrants was 30.9% and 20.1%, respectively [28]. Similarly, the prevalence of overweight among Ghanaian men in Australia was 53.3%, and among Ghanaian women it was 40.1% [30]. In The Netherlands, Germany, and the UK, the prevalence of overweight/obesity among Ghanaian immigrants ranged from 29.1% to 83.0% [16,51]. In Israel, the lowest reported prevalence of obesity among Ethiopian immigrants was 11%, and the combined prevalence of overweight/obesity was 42% [49]. Several studies examined the prevalence of overweight/obesity among African immigrants residing in the US. Across all studies conducted in the US, the lowest observed prevalence of overweight/obesity was 13% among Ethiopian immigrants [29], and the highest observed prevalence was 89% among Nigerian immigrants (overweight/obesity combined) [21].

#### 3.2.4. Smoking Prevalence

The prevalence of smoking was reported in 14/35 reviewed studies [16,17,18,19,24,26,28,29,32,33,35,40,42,45,49]. The immigrants were born in Ghana, Equatorial Guinea, Ethiopia, Sudan, and Somalia. The pooled prevalence of smoking was 7% (Figure 6), and the total prevalence ranged from 0.7% [16] to 14.8% [35]. When stratified by gender, the prevalence of smoking was lower among women (range: 0% [32] to 9.4%) [42] than men (range: 0.8% to 29.8%) [42]. Similar to hypertension and diabetes, smoking prevalence differed among immigrant groups based on the host country. Among Ghanaian immigrants in The Netherlands, the prevalence ranged from 0.8% [19] to 18.3% [42] among men and 1.0 [19] to 6.7 [42] among women. In comparison, Dutch men and women had a comparable smoking prevalence of 27% and 24.2%, respectively [18]. In Germany (Berlin), the total range varied between 9.3% [16] and 14.8% [35]. In contrast, in the UK (London), the range varied between 0.7% [16] and 1.4% [35].

Somali immigrants had a smoking prevalence of 1.98% [33] among women and 14.6% [33] among men in the US, 0% among women [32,40] and 4.0 to 4.1% [32,40] among men in Finland, and 0% among women and 19% among men [26] in Norway. For host comparators, the smoking prevalence among the Finnish population was higher than among Somali immigrants, averaging 20.4% among men and 27.9% among women in Finland (Appendix A).

#### 3.2.5. Dyslipidemia Prevalence

The prevalence of dyslipidemia among African immigrants who were born in Ethiopia, Ghana, Nigeria, Somalia, or Zimbabwe was reported in 9/35 of the studies. Studies evaluated the prevalence of high triglycerides, high total cholesterol, high LDL cholesterol, and low HDL cholesterol. Dyslipidemia prevalence ranged from 11 to 77.2% (Figure 7). We observed gender differences in the prevalence of dyslipidemia with mixed findings. The prevalence of hypercholesterolemia was higher among Zimbabwean men (26.1%) than women (17.8%) in the US [23]. However, among Somali men and women in Finland, there was no difference in the prevalence of dyslipidemia: 77.1% and 77.2%, respectively [32]. Yet, Somalian women (62.5%) in the US had a higher prevalence of hypercholesterolemia than Somalian men (49%) [33]. Among Ethiopian immigrants in Minnesota, US, there was a non-significant difference in the prevalence of hypercholesterolemia (≥240 mg/dL) between male and female immigrants [34]. Among Ghanaian immigrants in Amsterdam, Berlin, and London, the prevalence of hypercholesterolemia was 11.3%, 12.3%, and 11.1% [16], respectively. The prevalence of low, high-density lipoprotein (HDL)-cholesterol was 20.3% among Somalian men and 43.2% among Somalian women in Finland [32].

#### 3.2.6. Kidney Disease and Cardiovascular Disease Prevalence

The prevalence of chronic kidney disease was examined in 1/35 studies. It reported on Ghanaian immigrants living in The Netherlands, Germany, and the UK. The prevalence was 9.1%, 11.9%, and 10.3%, respectively [16]. Cardiovascular Disease (CVD) prevalence among African immigrants was reported in 3/35 [23,29,33] studies. African immigrants were from Somalia, Zimbabwe, and Ethiopia. There were gender differences in the prevalence of CVD. CVD prevalence among Zimbabwean women (6.1%) was higher than among men (4.8%) in the US [23]. The prevalence rates of ischemic heart disease, peripheral artery disease, and cerebral vascular accidents were 1.7%, 0.5%, and 4.3% [29], respectively. Native-born Israeli adults had a reported prevalence of 5.4%, 1.2%, and 2.5%, respectively [29].

## 4. Discussion

We conducted this systematic review to examine the prevalence of CMD risk factors among SSA immigrants residing in HICs. We observed heterogeneity in the burden of CMD risk factors by African country of origin and HIC country of residence. The most prevalent CMD risk factors were hypertension, overweight/obesity, and diabetes. We observed gender differences in the prevalence of CMD risk factors. The prevalence of hypertension, diabetes, dyslipidemia, and tobacco use was higher among men, except for overweight/obesity, which was higher among women.

We found a higher prevalence of hypertension among African immigrants from HICs than the global adult hypertension prevalence of 28.5% in HICs (2010) [52]. This is consistent with the trend of hypertension prevalence increasing in LMICs compared to HICs. Most studies examined the prevalence of African immigrants residing in Europe, while fewer were conducted in North America. In the study by Agyemang et al. conducted in Amsterdam, the prevalence of hypertension was significantly higher among Ghanaian males (61.6%) and females (50.9%) than Dutch males (33.7%) and females (18.9%) [18]. A prior seminal study by Cooper and colleagues has shown a gradient of hypertension risk among West African descent populations residing in SSA, the Caribbean Islands, and the US [53]. The higher prevalence of hypertension among African immigrants in HICs compared to the host residents may include higher levels of obesity, dietary changes associated with acculturation, and difference in leisure-time physical activity levels [54,55]. Longer residence in HICs has been shown to increase the prevalence of hypertension among diverse immigrant groups [56,57,58]. Sex differences in the prevalence of hypertension are well-established and were confirmed among African immigrants in this review. The reasons for the higher prevalence of hypertension among men may include the biological factors, differences in sex hormones and activation of the renin-angiotensin system [59], and behavioral factors such as higher intake of alcohol [60], which is a risk factor for hypertension.

We observed an alarming prevalence of overweight/obesity among African immigrants residing in HICs. The highest rates of overweight/obesity were observed among Nigerian immigrants living in the US. The rates of overweight/obesity mimic the rising rates of overweight/obesity in SSA, where obesity rates are expected to have tripled by 2030 [61]. Sex differences in the prevalence of overweight/obesity were pronounced, with females having a much higher burden. In the RODAM (Research for Obesity and Diabetes among African Migrants) Study, the prevalence of obesity among Ghanaian women (49.4%) was almost 2.6 times that of Ghanaian men (18.8%) in Amsterdam [35]. Cultural preferences may explain the high prevalence of obesity among African women for larger body size, which indicates beauty, fertility, health, and wealth [62]. This preference contrasts with the European perspective, where obesity is associated with illness [63]. The migration of Africans from SSA to obesogenic environments in HICs, which are characterized by easier access to high-calorie fast foods, may further worsen obesity among Africans [64]. Although the use of BMI, a metric derived in European populations [65], as a measure of cardiometabolic health in Africans represents a limitation of several studies, the relationship between obesity and mortality is well known [66,67].

Globally, the prevalence of diabetes has increased to epidemic proportions. The pooled prevalence of diabetes in this review was 11%, the lowest was among Somalians in the US (4.5%) [31] and the highest among Ethiopians in Israel (17.4%) [29]. The International Diabetes Federation has projected that the African region will experience the largest increase (129%) in diabetes prevalence, from 24 million in 2021 to 55 million by 2045 [68]. While the pathogenesis of diabetes in SSA is poorly understood, some studies have shown that the phenotype of diabetes in Africa is distinct, with most Africans developing diabetes at a younger age (<50 years), at a relatively leaner body size (normal BMI), and with beta-cell secretory dysfunction [69]. Ishimwe and colleagues have shown that beta-cell failure rather than insulin resistance may be the underlying cause of abnormal glucose tolerance among African immigrants [70]. Since obesity and smoking are associated with beta-cell dysfunction, [71] interventions targeting these risk factors, which are more prevalent in HICs, may prevent CMD among African immigrants [61].

Overall, this review highlights a high prevalence of CMD risk factors among SSA immigrants living in HICs. To address this global health challenge, it is critical to examine barriers to preventive health care access and utilization in this population. In countries such as the US, health insurance coverage has been identified as a major barrier to health care access and utilization. Previous studies have reported lower insurance rates in this group [72] and restricted access to public assistance and health benefits. In some countries in Europe, restricted legal entitlement to health services remains a significant barrier to preventive care and cardiometabolic screening and care [73]. Other obstacles to utilization that have been previously identified include limited health literacy, unfamiliarity with the health system of the host country, social exclusion, fear and low knowledge of chronic disease etiology and prevention, provider communication barriers, distrust of the health system, negative associations and perception towards health-seeking, etc. [73,74]. The higher prevalence of CMD risk factors and disease among SSA immigrants can be reversed by designing and executing health programs to improve the cardiometabolic health of African immigrants. There is an urgent need for health policy reforms entrenched in health and social justice to integrate African immigrants into health systems and improve access, utilization, and preventive care outcomes [75].

This review has some limitations. First, we excluded articles that aggregated data on different African immigrant groups and did not report data for specific African countries of origin. We selected this approach because we wanted to identify the burden of CMD among specific African immigrant groups and not assume that the African immigrant population was homogenous. Second, due to high heterogeneity in the studies, we could not conduct a meta-analysis of effect sizes; hence, a narrative summary approach was adopted for this synthesis. Additionally, only one study examined the prevalence of CKD and thus these results cannot be generalized. Lastly, studies not published in English or French may have been missed. However, this study has some strengths. To our knowledge, this is the first detailed systematic review of the cardiometabolic health of SSA immigrants across HICs. The review included a comprehensive search of multiple electronic databases. In addition, a meta-analysis of proportions was conducted to derive pooled prevalence for each cardiometabolic risk factor.

## 5. Conclusions

We observed a substantial burden of cardiometabolic risk factors such as hypertension, overweight/obesity, and dyslipidemia among African immigrants residing in HICs. These findings highlight the critical need for investment in preventive approaches to improve the cardiometabolic health of SSA immigrants. Policies to enable improved access to care and effective health programs are crucial to enhancing the cardiometabolic health outcomes of sub-Saharan African immigrants as they settle into new lives in their host countries.

## Figures and Tables

**Figure 1 ijerph-19-07959-f001:**
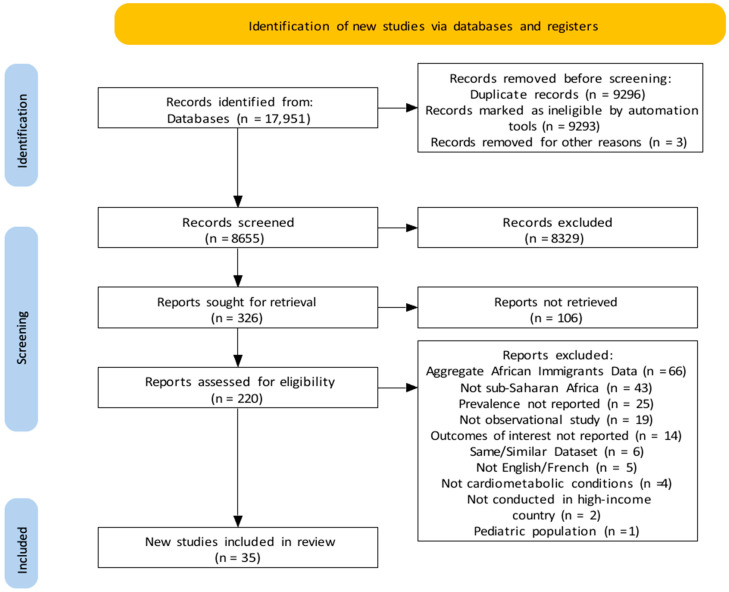
PRISMA flow diagram showing literature search and article inclusion [15].

**Figure 2 ijerph-19-07959-f002:**
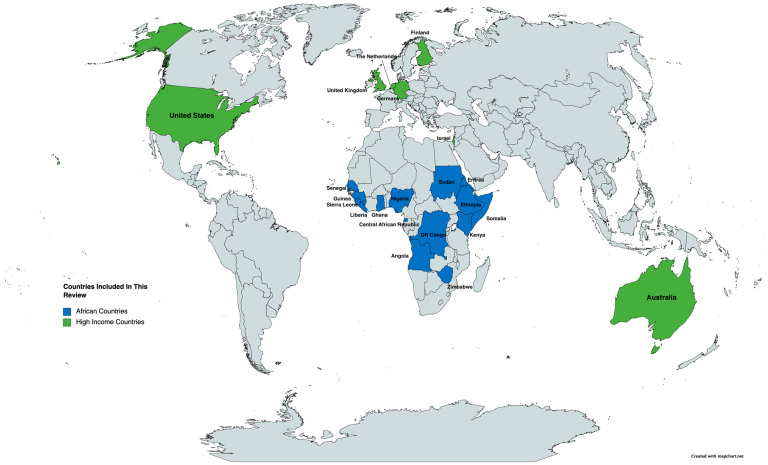
African Countries of Origin. Note, Somaliland is not distinguished from Somalia.

**Figure 3 ijerph-19-07959-f003:**
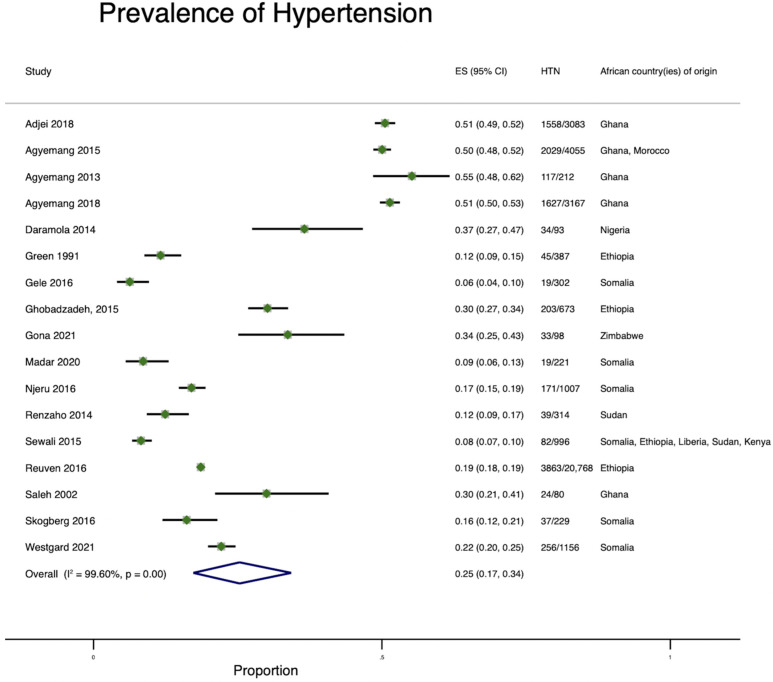
Pooled prevalence of hypertension among African immigrants residing in high-income countries.

**Figure 4 ijerph-19-07959-f004:**
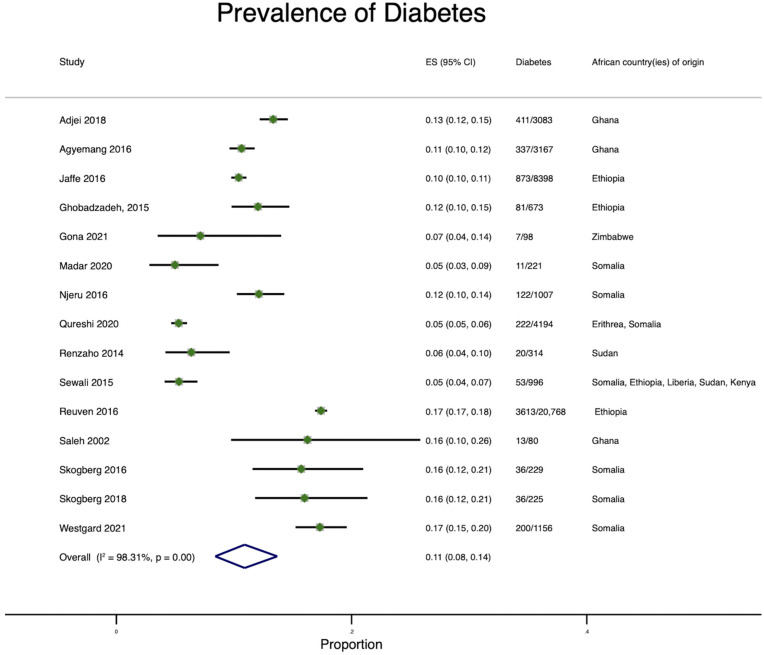
Pooled Prevalence of diabetes among African immigrants residing in high-income countries.

**Figure 5 ijerph-19-07959-f005:**
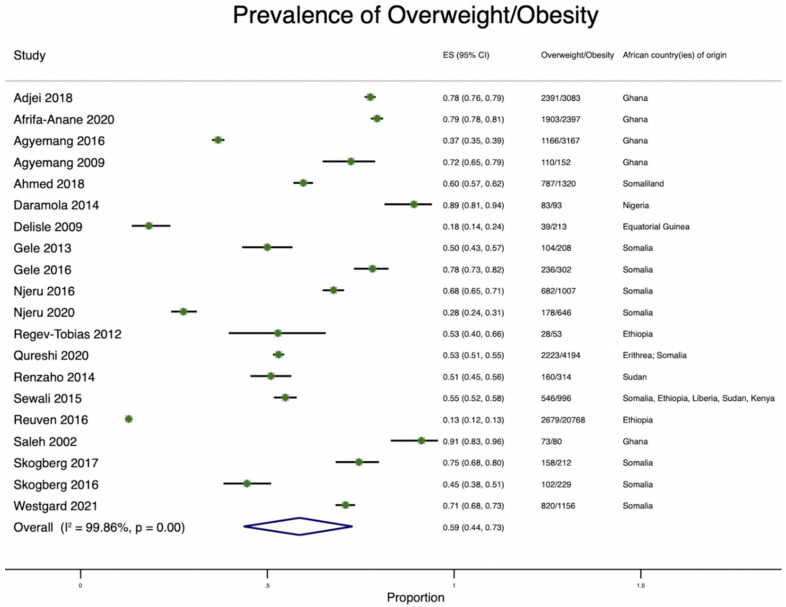
Pooled Prevalence of overweight/obesity among African immigrants residing in high-income countries.

**Figure 6 ijerph-19-07959-f006:**
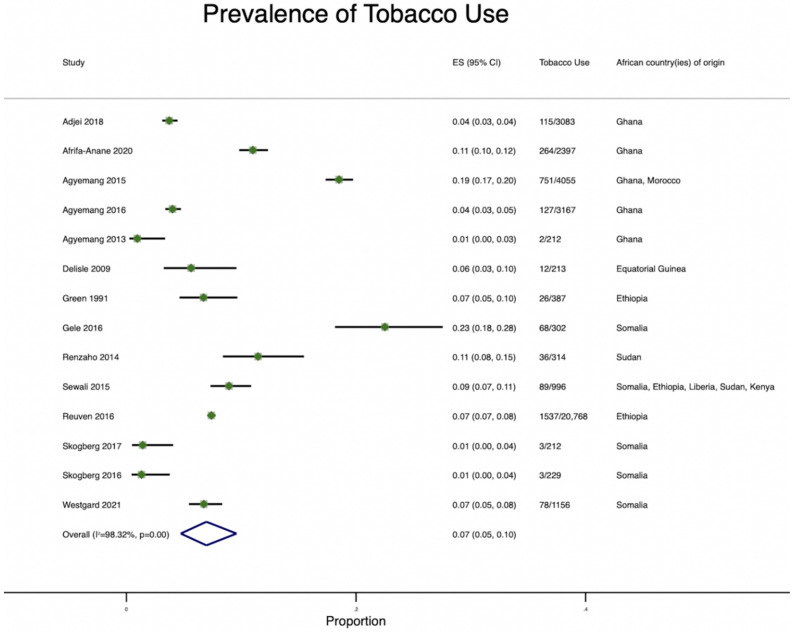
Pooled Prevalence of tobacco use among African immigrants residing in high-income countries.

**Figure 7 ijerph-19-07959-f007:**
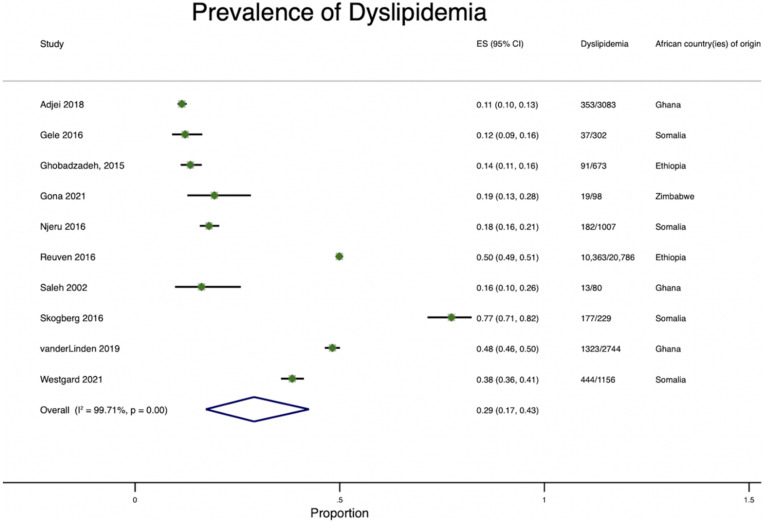
Pooled Prevalence of dyslipidemia among African immigrants residing in high-income countries.

**Table 1 ijerph-19-07959-t001:** Studies examining the cardiometabolic health of African immigrants in high-income countries (N = 35).

First Author and Year	High Income Country	Country of Origin	Risk Factors							Study Design	Sample Size (N = Total Population n = African Immigrant Population)	Mean Age or Range (Years)	Comparison Population
			Hypertension	Diabetes	Obesity	Smoking	Dyslipidemia	CKD	CVD				
Adjei 2018	The Netherlands; Germany; UK	Ghana	*	*	*	*	*	*		Cross-sectional	N = 5607, n = 3083	44.4–47.5	N/A
Afrifa-Anane 2020	The Netherlands; Germany; UK	Ghana			*	*				Cross-sectional	N = 4760, n = 2397	Men: 46.83, Women: 45.80	N/A
Agyei 2014	The Netherlands	Ghana	*		*	*				Cross-sectional	N = 221, n = NR	44.6	N/A
Agyemang 2015	The Netherlands	Ghana	*			*				Cross-sectional	N = 12974, n = 1871	Men: 47.1, Women: 43.9	Dutch
Agyemang 2016	The Netherlands; Germany; UK	Ghana		*	*	*				Cross-sectional	N = 5659, n = 1316	Men 45.8-48.4, 44.7-47.7	N/A
Agyemang 2013	The Netherlands	Ghana	*			*				Cross-sectional	N = 212, n = NR	44.6	N/A
Agyemang 2018v b	The Netherlands	Ghana	*							Cross-sectional	N = 5659, n = 3167	Men 45.8- 48.4, Women 44.7-47.7	N/A
Agyemang 2009	The Netherlands	Ghana			*					Cross-sectional	N = 1471, n = 152	Men 40.1 Women 37.9	N/A
Ahmed 2018	Norway	Somaliland			*					Cross-sectional	N = 1330, n = 220	Men 39.7, Women 37.7	N/A
Daramola 2014	United States	Nigeria	*		*					Cross-sectional	N = 129, n = 91	48.8	African Americans
Delisle 2009	Spain	Equatorial Guinea			*	*				Cross-sectional	N = 213, n = NR	Male: 33.2, Women: 36.5	N/A
Gualdi-Russo 2009	Italy	Senegal	*		*					Cross-sectional	N = 401, n = 44	17–65	Italians
Guerin 2007	New Zealand	Somalia			*					Cross-sectional	N = 314, n = NR	12–66	New Zealanders
Gele 2013	Norway	Somali			*					Cross-sectional	N = 208, n = NR	≥25	N/A
Gele 2016	Norway	Somalia	*		*					Cross-sectional	N = 302, n = NR	36.13	N/A
Jaffe 2016	Israel	Ethiopia		*						Cohort	N = 24,375, n = 7994	N = 40.3	Israeli
Ghobadzadeh, 2015	United States	Ethiopia		*				*		Cross-sectional	N = 718, n = NR	≥18	N/A
Gona 2021	United States	Zimbabwe	*	*	*		*		*	Cross-sectional	N = 98, n = NR	47.5	N/A
Goosen 2014	The Netherlands	Angola, Burundi; Democratic Republic of Congo; Guinea; Sierra Leone; Somalia; Sudan		*						Cross-sectional	N = 1255, n = 693	20–79	N/A
Madar 2020	Norway	Somalia								Cross-sectional	N = 221, n = NR	39	N/A
Njeru 2016	United States	Somalia	*	*	*		*			Cohort	N = 2017, n = 1007	≥17	N/A
Njeru 2020	United States	Somalia			*					Cross-sectional	N = 646, n = NR	37.5	N/A
Obisesan 2017	United States	Nigeria			*					Cross-sectional	N = 181, n = NR	NR	N/A
Regev-Tobias 2012	Israel	Ethiopia			*					Cross-sectional	N = 53, n = NR	32.3	N/A
Qureshi 2020	Norway	Eritrea; Somali			*					Cross-sectional	N = 4194, n = 344	45–66	N/A
Renzaho 2014	Australia	Sudan	*	*	*	*				Cross-sectional	N = 314, n = NR	18–70	N/A
Sewali 2015	United States	Somalia; Ethiopia; Liberia; Sudan; Kenya,	*	*	*					Cross-sectional	N = 996, n = NR	35	N/A
Reuven 2016	Israel	Ethiopia	*	*	*	*	*		*	Retrospective Cohort	N = 58,901, n = 20,768	51	Israelis
Saleh 2002	Australia	Ghana	*		*					Cross-sectional	N = 80, n = NR	Men: 40.4, Women: 34.8	N/A
Torp 2015	Sweden	Somali			*					Cross-sectional	N = 114, n = NR	34.8	N/A
Skogberg 2017	Finland	Somalia	*	*	*	*	*			Cross-sectional	N = 1632, n = 212	Men: 40.5, Women 42.3	Finnish
Skogberg 2016	Finland	Somalia	*	*	*	*	*			Cross-sectional	N = 1813, n = 229	18-64	Finnish
Skogberg 2018	Finland	Somalia		*						Cross-sectional	N = 1804, n = 225	Men: 31.1; Women: 35.3	Finnish
vanderLinden 2019	The Netherlands; Germany; UK	Ghana			*		*			Cross-sectional	N = 5482, n = 2744	25–70	N/A
Westgard 2021	United States	Somalia	*	*	*	*	*		*	Cross-sectional	N = 1156, n = NR	≥18	N/A

CVD, Cardiovascular Diseases; CKD, Chronic Kidney Disease; UK: United Kingdom; NR = Not Reported; NA = Not Applicable; * Risk Factor or Condition Present.

## Data Availability

Not applicable.

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
