# Peer review of "The Cardiometabolic Health of African Immigrants in High-Income Countries: A Systematic Review"

_ijerph, 2022, doi:10.3390/ijerph19137959_

Round 1

Reviewer 1 Report

Thank you for the opportunity to review this systematic review on examining the prevalence of cardiometabolic disease risk factors among African immigrants who reside in high income countries. Some of my comments/questions as follows:

1) First paragraph of the introduction talks about cardiometabolic diseases and how the prevalences of these diseases can be affected by social determinants of health. But when I read the objective of the systematic review, it talks about cardiometabolic risk factors which I guess, is slightly different from cardiometabolic diseases themselves. So the authors may want to elaborate on these points better.

2) I'm surprised that there are no mention of Asia in high income countries. What I know, there are large group of African immigrants in China and other parts of Asia. 

3) I noted that the search of literature ends around a year ago, in July 2021. Any chance that there will be new literature being published since the date of search as these may affect the findings of this review?

4) I noticed that there is a big team of reviewers for initial abstract and title screening. Any particular reason behind this? Just wondering whether the team has agreed on certain consensus of the articles wanted for this review as large number of reviewers may introduce biases.

5) I believe there is an updated version of PRISMA flow diagram (2020) that the authors can use for this review. 

6) Figures 3-7 - there is a high heterogeneity amongst the studies to perform the pooled prevalence. What are the authors' thought for this in terms of validity of the findings for this review?

7) What are the authors' plans after this review? What future studies or policies can be implemented from these findings?

Author Response

Response 1: Thank you for pointing out this inconsistency. We have revised the introduction section to make this point clearer. We sought to examine the prevalence of CMD and associated risk factors.

Response 2: This review was limited to sub-Saharan African countries residing in high-income countries. The World Bank designation of high-income countries excludes China (upper middle income) and other Asian countries which are also considered LMICs. Although there is a growing population of African immigrants in Asia, exploring the cardiometabolic health of African immigrants in other LMICs is beyond the scope of this review.

Reference

The World Bank. The World by Income and Region. https://datatopics.worldbank.org/world-development-indicators/the-world-by-income-and-region.html. Accessed June 20, 2022.

Response 3: We updated the review to May 2022 and found 2 articles that met our criteria.

Thank you for this suggestion. We conducted a preliminary search of the databases, using dates limits between our last search dates till date. This preliminary search and screening identified only 220 articles that may potentially meet the inclusion criteria for this review. Based on the state of the literature on this subject, it is unlikely that there will be significant changes to our results and conclusions with the inclusion of these three additional studies.

We intend to conduct an updated review that include Sub-Saharan immigrants from other WHO income level designation. This future work will strongly emphasize effectiveness of adapted interventions to improve cardiometabolic health in this population.

Response 4: Due to the large number of articles which were screened, we assembled a team that used the standardized process in COVIDENCE to minimize variability and standardize the initial abstract and title screening process. This approach allowed for minimization of article selection bias. In addition, the large number of reviewers allowed for independent screening of each article by at least two reviewers, and designation of an independent reviewer to serve as adjudicator.

Response 5: Thank you, we used the PRISMA flow diagram (2020) for this review and it has been cited in the paper

Response 6: Thank you for this comment. We did not perform a comparative meta-analysis, rather a meta-proportions or pooled summary of proportions to describe the prevalence reported in the reviewed articles (figure 3-7). The measure of heterogeneity I2 in meta-proportions is expected to be interpreted conservatively unlike its interpretations in comparative meta-analyses. High heterogeneity is sometimes expected in meta-proportions from prevalence studies due to variability in the place and time the studies were conducted particular in our review, which accounted for studies conducted over multiple decades and in various high-income countries across various continents.

Reference

Barker TH, Migliavaca CB, Stein C, Colpani V, Falavigna M, Aromataris E, Munn Z. Conducting proportional meta-analysis in different types of systematic reviews: a guide for synthesisers of evidence. BMC Med Res Methodol. 2021 Sep 20;21(1):189. doi: 10.1186/s12874-021-01381-z. PMID: 34544368; PMCID: PMC8451728.

Response 7: In the discussion section, we outlined the research and policy implications of our findings. For instance, we noted that: “The higher prevalence of CMD risk factors and disease among SSA immigrants can be reversed by designing and executing health programs to improve the cardiometabolic health of African immigrants. There is an urgent need for health policy re-forms entrenched in health and social justice to integrate African immigrants into health systems and improve access, utilization, and preventive care outcomes.[75]”

We believe our review highlights urgent need for addressing cardiometabolic risks in this population. In particular, through adapting effective interventions and engaging the Sub-Saharan African immigrant community in developing culturally appropriate, and inclusive programs targeted at improving cardiometabolic health.

Reviewer 2 Report

The review tilted "The Cardiometabolic Health of African Immigrants in 2 High-Income Countries: A Systematic Review" made by Danielle Mensah et all examined the prevalence of cardiometabolic risk factors among sub-Saharan African immigrants residing high-income countries. Recommendations from the Preferred Reporting Items for Systematic Reviews and Meta-Analysis (PRISMA) were used to conduct a systematic review.

The review evaluates an interesting and worthy of evaluation topic but before publication in my opinion it needs some corrections.

Here are my minor questions and comments:

Introduction section

1.     The term “chronic renal failure” should be changed to “chronic kidney disease” while the Authors use CKD abbreviation later, it also should be written everywhere where the kidney is mentioned.

Material and methods section

1.     The abbreviation SSA should be explained as it was mentioned here for the very first time.

2.     Used keywords in MeSH should be listed.

Kidney disease and cardiovascular section

1.     CHD abbreviation should be explained.

2.     “Peripheral artery disease” is not the same as “peripheral vascular disease” so the Authors should be more precise which disease is discussed.

3.     I am not sure if the prevalence of CKD should be described here while it was analysed only in one study.

Discussion section

1.     Line nr - 307 - CMD should be except CVD.

2.     Line nr - 337 - RODAM abbreviation should be explained.

Author Response

Comment 1: The term “chronic renal failure” should be changed to “chronic kidney disease” while the Authors use CKD abbreviation later, it also should be written everywhere where the kidney is mentioned.

Response 1: Thank you for pointing out this error. We have made this correction.

Material and methods section

Response 1: Thank you. We have introduced SSA (Sub-Saharan Africa) in the introduction section.

Response 2: The keywords are listed in the supplement (Table S2)

Kidney disease and cardiovascular section response

1.  We have modified this sentence for more clarity

2. We have clarified this in the manuscript, the disease being referenced is peripheral artery disease

3. We included the study reporting on CKD because this fit our inclusion criteria, but also to highlight the dearth in the literature on prevalence of CKD in the Sub-Saharan African immigrant population. Considering the increase in CKD prevalence in Sub-Saharan Africa, we believe the burden of CKD in Sub-Saharan African immigrant population is understudied and our review further reflect this gap in the literature.

Reference

  1. George JA, Brandenburg JT, Fabian J, Crowther NJ, Agongo G, Alberts M, Ali S, Asiki G, Boua PR, Gómez-Olivé FX, Mashinya F, Micklesfield L, Mohamed SF, Mukomana F, Norris SA, Oduro AR, Soo C, Sorgho H, Wade A, Naicker S, Ramsay M; AWI-Gen and the H3Africa Consortium. Kidney damage and associated risk factors in rural and urban sub-Saharan Africa (AWI-Gen): a cross-sectional population study. Lancet Glob Health. 2019 Dec;7(12):e1632-e1643. doi: 10.1016/S2214-109X(19)30443-7. PMID: 31708144; PMCID: PMC7033368.
  2. Kaze AD, Ilori T, Jaar BG, Echouffo-Tcheugui JB. Burden of chronic kidney disease on the African continent: a systematic review and meta-analysis. BMC Nephrol. 2018 Jun 1;19(1):125. doi: 10.1186/s12882-018-0930-5. PMID: 29859046; PMCID: PMC5984759.

Discussion section Response

  1. Thank you. We believe the reviewer meant for us to consistently use “CMD” instead of “CVD”. The manuscript has been updated to reflect this change.
  2. Thank you. We have defined the RODAM abbreviation